# Comparative Study of Pain-Related Responses of Male Piglets up to Seven Days of Age to the Application of Different Local Anaesthetics and Subsequent Castration

**DOI:** 10.3390/ani12202833

**Published:** 2022-10-19

**Authors:** Franz Josef Söbbeler, Sören Wendt, Andreas Briese, Julia Tünsmeier, Karl-Heinz Waldmann, Sabine Beate Rita Kästner, Alexandra von Altrock

**Affiliations:** 1Clinic for Small Animals, University of Veterinary Medicine Hannover, Hannover, Foundation, 30559 Hannover, Germany; 2Clinic for Swine, Small Ruminants, Forensic Medicine and Ambulatory Service, University of Veterinary Medicine Hannover, Foundation, 30173 Hannover, Germany; 3eduToolbox, 31157 Sarstedt, Germany

**Keywords:** castration, piglet, local anaesthesia, pain assessment, heart rate variability (HRV), lidocaine, procaine, mepivacaine, minimal alveolar concentration (MAC), isoflurane

## Abstract

**Simple Summary:**

Since 2021, surgical piglet castration must be performed with complete pain elimination according to the Animal Protection Law in Germany. General anaesthesia by isoflurane inhalation, which can be performed by the farmer, or by injection of ketamine and azaperone, which must be performed by a veterinarian, are the options available. At present, local anaesthesia is still under debate because of the lack of proof of complete pain elimination and the pain on injection. We tested three local anaesthetics (procaine, lidocaine, and mepivacaine) at two different doses each. Because pain responses can be masked by reactions caused by handling, the piglets were given superficial isoflurane anaesthesia. The pain on injection to the testes was compared with intramuscular injection, and the effectiveness during castration was compared among the local anaesthetics. Nocifensive movements, respiratory rate, blood pressure, heart rate and its variability as well as electroencephalography (EEG) changes were studied in relation to the painful interventions. Most indicators of nociception point to testicular injection pain being beyond intramuscular injection pain when an effective amount of local anaesthetic was used. However, complete pain elimination could not be achieved during castration under local anaesthesia.

**Abstract:**

To evaluate pain responses to intratesticular and subscrotal injection of three local anaesthetics and their efficacy during castration a randomized controlled study was conducted. In groups of 20 piglets, procaine (2%), lidocaine (2%), or mepivacaine (2%) were administered subscrotal and intratesticularly in two different dosages: 0.5 mL of the original substances or the maximum recommended dosage according to body weight diluted with isotonic saline to a volume of 0.3 mL per each injection site. Two placebo groups received the equivalent volume of isotonic saline. A control group was injected intramuscularly with 0.5 mL isotonic saline for injection pain comparison. Electroencephalographic changes, respiratory rate, heart rate and its variability, blood pressure, and nocifensive movements were assessed in superficial isoflurane anaesthesia. While EEG-changes and linear measures of heart rate variability did not appear conclusive, the low frequency/high frequency (LF/HF) ratio corresponded best with the other pain indicators recorded. The injection of 0.3 mL diluted local anaesthetic per injection site elicited significant fewer signs of pain compared to intramuscular injection of saline. However, pain reduction, but not complete pain elimination, during castration could only be achieved with 0.5 mL of the 2% local anaesthetics per injection site, whereby lidocaine and mepivacaine were the most effective.

## 1. Introduction

Surgical castration of male piglets is a painful procedure. Primarily, castration prevents boar taint, which affects the consumer acceptability of pork and pork products. Therefore, castration is not a producer’s decision but a market-driven choice [1]. In 2010, representatives of European farmers, the meat industry, retailers, scientists, veterinarians and animal welfare organisations voluntarily committed to abandoning piglet castration as of 1 January 2018 [1]. While surgical castration is still practiced in many EU countries, anaesthesia becomes more and more mandatory. Since 1 January 2021, in Germany, the surgical castration of male pigs aged less than eight days is no longer allowed without effective pain elimination (German Animal Protection Law (Tierschutzgesetz)). In comparable regulations of other European Union (EU) Member States, pain attenuation is considered as sufficient [2]. Anaesthesia for piglets can be either general or local. In Germany, general anaesthesia in piglets may be performed by inhalation of isoflurane or by intramuscular (i.m.) injection of ketamine in combination with the neuroleptic drug azaperone. Ketamine-azaperone anaesthesia must be carried out by a veterinarian. The main disadvantage of this technique is the prolonged recovery period, during which the piglets must be separated to prevent them from being crushed by the sow. Since 2020 anaesthesia with isoflurane for castration of piglets can be performed by the farmer and permits a safe and rapid anaesthetic induction as well as a brief recovery [3]. Inhalation anaesthesia with isoflurane by the farmer requires an anaesthetic device, which delivers isoflurane over a set period of time. Due to the lack of individual adjustment, sufficient depth of anaesthesia is not always achieved [4,5]. Additionally, isoflurane is a greenhouse gas, affecting the environment and the operator. For economic and animal welfare aspects anaesthesia for routine piglet castration should meet the following criteria: short induction and recovery period, quick and easy to be performed, cause minimal stress, cost-effective, no residuals, and a large therapeutic range of used drugs [6,7]. Therefore, local anaesthesia seems to be a good option to eliminate pain during castration. Topical anaesthesia has failed to provide the required analgesia for all surgical steps of castration [8], the effect of infiltration local anaesthesia in connection with piglet castration is the subject of controversial discussion. Local anaesthetics reversibly block voltage gated sodium channels inhibiting neural conduction. According to their structure, they are categorized into esters or amides, and into short- (e.g., procaine), intermediate (e.g., lidocaine, mepivacaine)—and long-acting (e.g., bupivacaine) compounds [9]. Depending on their p*K*a local anaesthetics differ in their onset of action, making lidocaine (p*K*a = 7.8) and mepivacaine (p*K*a = 7.9) faster acting than procaine (p*K*a = 9.0) [10]. Procaine has been assessed by the European Medicines Agency (EMA) as a local anaesthetic, which can be used in food producing animals and is currently the only substance approved for pigs in Germany. Lidocaine has been licensed in swine for cutaneous and epilesional use since November 16, 2020 [11]. However, injectable lidocaine and mepivacaine can only be used for piglets in accordance with the cascade rule. Lidocaine is up to twice as effective as procaine [12]. Studies on the efficacy of both substances during castration of piglets have shown a pain reducing effect [6,13,14,15,16,17]. The injection itself was repeatedly found to result in pain reactions [18,19,20,21]. Direct comparison of study results is almost impossible, as the injection site, injected volume and drug concentration differ, as do the variables used for pain detection. In two recent studies the effect of four local anaesthetics (4% procaine, 2% lidocaine, 0.5% bupivacaine, 2% mepivacaine) was compared [18,22]. It was shown that all four local anaesthetics reduced signs of nociception during castration, but it was also demonstrated that the intratesticular injection caused visible nociception indicated by increased limb movements [18]. The evaluation of nocifensive movements for pain recognition is useful in severe acute events [23]. Since nociceptive withdrawal responses can vary between pigs, it is reasonable to look at behaviour and physiology in combination to fully assess the impact of a painful event on the individual [23]. Parameters, like respiratory rate (RR) and heart rate (HR), blood pressure (BP) and processed EEG variables like Narcotrend-Index (NI), total power (PTOT), median frequency (MF) and 95% spectral edge frequency (SEF95), are objectively measurable and most of the parameters mentioned were already used in numerous studies to identify pain reactions in pigs [6,15,18,21,24,25]. Physiological responses of HR and BP result from activation of the autonomous nervous system and may be triggered by stress from handling and restraining [26]. To associate physiological reactions with nociception during castration general anaesthesia models can be used [6,18].

Heart rate variability (HRV) is a promising clinical tool to evaluate the autonomic response of different stressors in pigs [27,28]. It indicates the variation in the time interval between consecutive heartbeats and represents the interplay of the parasympathetic nerves, which slow HR, and the sympathetic nerves, which accelerate it. A recent systematic review concluded, that HRV is a good measure of autonomic reactivity to nociceptive stimulation in man [29]. Investigations into HRV to prove pain reactions in pig castration were already conducted [30,31].

The main aim of the study was to assess the suitability of local anaesthesia for piglet castration under 8 days of age fulfilling the requirements of the German Animal Protection Law with respect to the castration procedure and pain of intratesticular and subscrotal injection. In order to reduce stress-induced responses caused by handling and restraining piglets were held in a standardized, minimum alveolar concentration (MAC)-based subanaesthetic isoflurane anaesthesia.

Our main hypothesis was that local anaesthesia with mepivacaine, lidocaine or procaine will lead to significant reduction in SEF 95% changes in response to castration. Secondly, we hypothesised that testicular injection leads to more nocifensive responses than i.m. injection and thirdly, local anaesthesia with lidocaine and mepivacaine leads to less nocifensive and autonomic responses to castration than procaine.

## 2. Materials and Methods

This study was reviewed and approved by the ethical committee for animal experimentation of the Federal State Office for Consumer Protection and Food Safety of Lower Saxony, Germany (33.9-42502-04-19/3218). All animal procedures were performed according to the German Animal Protection Law (Tierschutzgesetz).

### 2.1. Animals

All piglets included in this study were male, two to seven days of age, and weighing one to three kg of bodyweight. They were acquired from four different producer farms (Main Study: Farm A: 41 piglets, Farm C: 88 piglets, Farm D: 51 piglets; no piglet from Farm B was included in the main study) in Lower Saxony, Germany, and have not been pretreated at the farm of origin. No piglet was subjected to teeth clipping or tail docking. Health status and descendance of both testicles into the scrotum was confirmed by a clinical examination. Piglets with hernia scrotalis or inguinalis were excluded. After completion of the study the piglets were raised motherless in an artificial rearing system (rescue deck).

Based on electroencephalogram (EEG) response to nociception data from Kulka, et al. [32], an a priori power calculation resulted in a sample size of 20 piglets per group to detect the absence of a change of 20% in SEF 95% with an effect size of 0.83, an alpha error of 5% and a power of 95% (G*Power 3.1.9.4; Heinrich Heine University, Düsseldorf, Germany).

### 2.2. Study Design and Procedure

#### 2.2.1. Pre-Trial

In a pre-trial, distribution of a local anaesthetic after intratesticular and intrafunicular injection was evaluated in ten piglets via computed tomography

Anaesthesia was induced via a face mask with 5 vol% isoflurane in 100% oxygen and a fresh gas flow of 4 L min^−1^. After reaching a sufficient depth of anaesthesia the piglets were orotracheally intubated with a Murphy style cuffed endotracheal Tube I.D. 2.5 mm (Rüschelit^®^ Super Safety Clear, 2.5 mm, Teleflex Medical Sdn. Bhd., Kamunting, Malaysia). According to positioning in commercial castration racks the piglets were fixed in supine position in a radiolucent foam positioning device, so that the testicles were easily accessible. As a surrogate for the corresponding volume of a local anaesthetic, a 1:7 dilution of a non-ionic, water-soluble X-ray contrast medium (iobitridol, Xenetix^®^ 300, Guerbet, Villepinte, France) with lidocaine hydrochloride (Lidor^®^ 20 mg/mL, WDT, Garbsen, Germany) was used in order to obtain a comparable viscosity and pH value as the pure local anaesthetic solution. At each testicular side a total volume of 0.3 mL was injected with a 25G × 5/8 needle (Neoject^®^, Dispomed Witt oHG, Gelnhausen, Germany). Five different injection techniques were investigated: intrafunicular (0.3 mL—palpation and fixation of the funiculus with one hand while injection was performed in an approx. 45° angle from caudal with the other hand), intratesticular (0.3 mL—fixation of the testicle with one hand while injection was performed in a 90° angle), subcutaneous in the subscrotal tissue (subscrotal) (0.3 mL—lifting a skinfold with one hand while injection was performed in an approx. 45° angle from caudal with the other hand), a combination of subscrotal (0.15 mL) and intrafunicular injection (0.15 mL) and a combination of subscrotal (0.15 mL) and intratesticular injection (0.15 mL) was performed. Injections were performed without aspiration prior to injection. Two animals were used per injection technique. A 64-multi-detector-row CT scanner (Phillips Brilliance 64, Philips GmbH, Hamburg, Germany) was used for image acquisition. CT scans with a slice thickness of 0.64 mm were acquired two minutes after injection and repeated 4, 6, 8 and 10 min after injection.

Acquired images were visually analysed for distribution of contrast media. For evaluation of temporal distribution, changes in distribution pattern over time were evaluated visually.

#### 2.2.2. Main Study

The main study was designed as a placebo controlled, blinded, randomized study with 9 parallel groups (Table 1). Randomization was performed with the aid of www.randomizer.org (accessed on 11 February 2020).

The three local anaesthetics procaine (Procamidor^®^ 20 mg/mL, WDT, Garbsen, Germany), lidocaine (Lidor^®^ 20 mg/mL, WDT, Garbsen, Germany) and mepivacaine (Mepidor^®^ 20 mg/mL, WDT, Garbsen, Germany) and the placebo (NaCl 0.9%, B.Braun Melsungen AG, Melsungen, Germany) were compared each in 2 different volumes and doses. According to the results of the preliminary investigation, each local anaesthetic or saline was administered at four locations (intratesticular left, subscrotal left, intratesticular right and subscrotal right) with a volume of either 0.5 mL or 0.3 mL resulting in a total volume of 2 mL or 1.2 mL per piglet. In groups where 0.5 mL were administered at each location undiluted local anaesthetic at a concentration of 2% was injected (procaine—group P_0_._5_, lidocaine—group L_0_._5_, mepivacaine—group M_0_._5_ and the placebo group with normal saline—group S_0_._5_). In groups where 0.3 mL were injected at each location the maximally recommended dose according to summary of product characteristics” (SPC) (https://vetidata.de/ accessed 11 February 2020) was calculated for each individual piglet and diluted with normal saline to a total volume of 1.2 mL (procaine—group P_0_._3_, lidocaine—group L_0_._3_, mepivacaine—group M_0_._3_ and placebo normal saline—group S_0_._3_). The maximally recommended doses were 5 mg kg^−1^ for procaine, 4 mg kg^−1^ for lidocaine and 0.4 mg kg^−1^ for mepivacaine. The dose for mepivacaine was calculated from a dose stated for horses. To compare the nociceptive response caused by intratesticular and subscrotal injection with a common clinical nociceptive stimulus one group was included that received an intramuscular injection of 0.5 mL normal saline (group IM).

##### Anaesthesia and Instrumentation

Anaesthesia was induced by mask with 5 vol% isoflurane (Isofluran Baxter^®^, Baxter Deutschland GmbH, Unterschleißheim, Germany) in 100% oxygen with a flow of 4 L min^− 1^. Upon loss of consciousness a 4 Fr feeding tube (Ernährungssonde CH 4, B.Braun Melsungen AG, Melsungen, Germany) was inserted into the ventral meatus nasi to determine inspiratory and expiratory isoflurane concentration and respiratory rate with the multigas module of a multiparameter anaesthesia monitor (Datex Ohmeda S5, GE Healthcare Finland OY, Helsinki, Finland). Isoflurane was reduced to maintain an Fe_ISO_ of 1 Vol% [representing 0.8 MAC determined in a similar population of piglets [33]. Two attempts were made to catheterize the saphenous artery with an over the needle catheter (Introcan^®^ 22G, B.Braun, Melsungen, Germany). Invasive measurement of arterial blood pressure and pulse rate was performed with the multiparameter anaesthesia monitor and a pressure transducer (Meritrans DTXPlus^®^ Disposable Transducer, Merit Medical GmbH, Eschborn, Germany) connected via fluid filled low compliance extension lines to the arterial catheter. No intravenous fluids were given. In case of unsuccessful catheterization, the procedure was continued without invasive measurement of blood pressure and pulse rate. Adhesive electrodes were attached to each leg to record a lead II electrocardiogram (Televet^®^ 100. Rösch & Associates Information engineering GmbH, Frankfurt am Main, Germany). Recording of the EEG signal was performed as described by Waldmann, et al. [21] with the Narcotrend^®^-Compact-Monitor Version 5.0. Finally, the piglets were placed in dorsal recumbency according to positioning in a castration cradle. Animals were warmed with the aid of a heating lamp to maintain body temperature above 38 °C. After instrumentation and an equilibration period of 10 min at a Fe_ISO_ of 1 Vol%, the injection of the test drugs was performed.

##### Injection Technique

In Group IM one intramuscular injection of 0.5 mL saline was performed with a 18G × 2” canula (BD Microlance 3, BD GmbH, Heidelberg, Germany) behind the base of the ear. The piglets in Group IM were recovered without castration. In all other groups half of the total injection volume was drawn up in a syringe, the scrotal skin was punctured, and half of the volume was administered intratesticularly and half subscrotal when the needle was withdrawn (0.5 mL or 0.3 mL of the solution per localisation). Injections were performed in each piglet always starting on the right side, followed by the left side. For intratesticular injection, the testicle was fixated with thumb and index finger of one hand and the injection was performed with a 25G × 5/8 needle (Neoject^®^, Dispomed Witt oHG, Gelnhausen, Germany) into the middle of the testicle without prior aspiration.

##### Castration

The castration was performed 5 to 15 min maximum after intratesticular and subscrotal injection, depending on the loss of sensitivity of the scrotal skin. Two incisions parallel to the raphe scroti were made and the testicles were exteriorized. Both testicles were removed at the same time by use of an emasculator. To ensure an adequate haemostasis compression of the emasculator was maintained for 20 s. Meloxicam 0.4 mg kg^−1^ (Melosolute^®^ 5 mg/mL, CP Pharma, Burgdorf, Germany) was administered at least 20 min before the surgical procedure i.m. After data acquisition, the piglets were recovered and placed in a nursing pen.

##### Data Recording

Nocifensive movements

Nocifensive movements during injection at each location were scored as (0)—no movement, (1)—minor purposeful movements (minor movements of one or 2 limbs) or (2)—major purposeful movements (more than 2 limbs or head). As any purposeful movement (movement associated directly related to the stimulus during injection or castration) was considered not conform with the Animal Protection Law, for statistical analysis only purposeful movement yes or no was differentiated. All scores for nocifensive movements were performed live by the same investigator blinded to treatment. One minute after injection followed by every 2 min sensitivity of the scrotal skin in the area where the injections were performed was examined. A pean clamp (Peha^®^ instrument Pean Klemme, Hartmann, Heidenheim Germany) equipped with a rubber hose -to avoid severe tissue trauma- was clamped to the first ratchet lock for 1 s. Castration was performed if no aversive response was noted, or a maximum of 15 min had passed. A period of 5 min after injection was always awaited to ensure an adequate time for onset of action of the injectate.

During castration nocifensive movements were scored as above for the timepoints “skin incision”, “exteriorization of the testicles” and “emasculation” which were performed with an interval of 15 s.

Respiratory rate and blood pressure

The following timepoints (Figure 1) were defined and analyzed: Baseline injection (BL-INJ) values were recorded every 15 s over a period of 2 min before injection and the mean was calculated. Baseline castration (BL-CA) value was as single value recorded directly before skin incision. As cardiorespiratory changes in response towards a short-lasting nociceptive stimulus only persist for a short period the highest value during a one-minute period after start of injection (INJ-Max) and during castration (CA-Max) were recorded. After castration, respiratory rate and mean arterial blood pressure were recorded every 15 s for 2 min (POST-CA) and the mean was calculated from the 8 individual values. A change of 10% in cardiorespiratory parameters compared to baseline was calculated for absolute values of respiratory rate, heart rate and MAP, as this amount of change would imply presence of nociception in a clinical setting.

Heart rate variability

An ECG was recorded with the Televet^®^ 100, Software Version 6.2.0 (Engel Engineering Services GmbH, Heusenstamm, Germany) and RR interval sections were extracted and the following time periods defined for analysis:

BL-INJ^e^ baseline 2 min before injection

INJ^e^—1 min after start of injection

BL-CA^e^” baseline 2 min before castration (skin incision)

CA^e^—castration, start of skin incision until 20 s after start of emasculation

POST-CA^e^ post castration, 2 min after “CA^e^”.

The superscript “^e”^ was added to differentiate these time periods/episodes from the time points for MAP and respiratory rate. This also refers to EEG data.

The RR-interval sections were transferred to Kubios^®^ RV version 2.0 (Biosignal Analysis and Medical Imaging Group, University of Kuopio, Kuopio, Finland). The ECG recordings were checked visually, errors were edited manually, and the data were analysed afterwards.

Time domain HRV analysis included mean heart rate, standard deviation of the heart rate (SDHR), mean RR-Interval (Mean RR) and the standard deviation of the RR-intervals (SDRR). The LF/HF ratio was calculated during frequency domain analysis with the low frequency band (LF) defined at 0.02–0.15 Hz and the high frequency band (HF) according to respiratory rate at 0.25–1.4 Hz.

EEG VariablesThe following variables were recorded and values for every 5 s time interval were exported: Narcotrend-Index (NI), total power (P_TOT_), median frequency (MF) and 95% spectral edge frequency (SEF_95_).Analysed timepoints (epochs) were:
BL-INJ^e^: baseline before injection for 2 minINJ^e^: 1 min after start of injectionBL-CA^e^: baseline 2 min before castration (skin incision)CA^e^: castration, start of skin incision until 20 s after start of emasculation.POST CA^e^: post castration, 2 min after the end of emasculation


### 2.3. Statistical Analysis

Data were analysed using R version 3.4.4. (The R Foundation for Statistical Computing, Vienna, Austria). The distribution of data was tested with Shapiro–Wilk Test and Histograms. Normally distributed data were analysed within groups with a paired t-test and an ANOVA for repeated measurements. Wilcoxon sign Rank test was used to compare nonparametric data within groups. In between groups an ANOVA for independent variables was used for normally distributed data and the Kruskal–Wallis-Test followed by Wilcoxon’s two sample test in case of significant differences was used to compare nonparametric data. Level of significance was set at 5%. Statistical analysis of ordinal data was performed with the Chi-square-test followed by Fisher’s exact test in case of cell values below 5.

## 3. Results

### 3.1. Pre-Trial

The 10 piglets used were between 2 and 7 days old and had a body weight between 1.9 and 2.4 kg.

Figure 2A-C show examples of the distribution of the surrogate within 2 min after injection at the respective sites. After intratesticular injection (*n* = 8) the surrogate was visible in the testis and along the spermatic cord into the abdominal cavity. Only in one piglet the local anaesthetic was falsely injected which resulted in a mainly intrascrotal distribution. Intrafunicular injection resulted in less consistent distribution in the spermatic cord, in the scrotum and in the abdominal cavity or in the subcutaneous tissue in the inguinal area. All subscrotal applications (*n* = 12) resulted in a subcutaneous depot.

In all localizations the distribution of the surrogate did not change over time.

### 3.2. Main Study

Mean age of the 170 piglets included in this study was 4 days with a mean bodyweight of 1.95 ± 0.31 kg.

#### 3.2.1. EEG

Baseline values for NI before injection (BL-INJ^e^) were comparable among all investigation groups (Figure 3). Changes in EEG parameters NI, P_TOT_, MF, SEF_95_ in response to injection or castration did not follow a consistent pattern and did not show any significant differences among the groups.

#### 3.2.2. Response to Injection

Nocifensive movements

Compared to intramuscular injection, intratesticular injection in group P_0_._5_ resulted in markedly stronger nocifensive movements (score 2). Intratesticular injection of the smaller volume of 0.3 mL led to statistically significantly less nocifensive movements than the administration of a volume of 0.5 mL even in comparison to the control group (Figure 4).

Cardiorespiratory changes

No significant differences were seen between BL-INJ and INJ-Max after injection of local anaesthetic in respiratory rate, heart rate and RR-intervals, but groups with larger volumes administered showed a higher difference compared to IM. Difference values in MAP were statistically significantly greater in P_0_._3_ and S_0_._5_ compared to IM. A 10% increase in respiratory rate was seen in all groups but IM and P_0_._3_ and in MAP for all treatments but IM. L_0_._3_ and M_0_._3_. There was no group with an increase in heart rate above 10%. For P_0_._5_ difference of SDHR was significantly higher compared to IM and difference of SDRR was higher compared to IM and M_0_._5_ whereas the LF/HF-ratio increased statistically higher during injection in group P_0_._5_ and S_0_._5_ compared to IM (Figure 5A–G).

#### 3.2.3. Response to Castration

Onset of action

Median (Minimum; Maximum) values of the time period between the intratesticular/subscrotal application and castration for groups P_0_._3_, P_0_._5_, L_0_._3_, L_0_._5_, M_0_._3_, M_0_._5_, S_0_._3_ and S_0_._5_ were 5 (5; 15), 5 (5; 7), 5 (5; 10), 5 (5; 5), 11 (5; 15), 5 (5; 5), 11 (5; 15), and 5 (5; 15) minutes, respectively (Figure 6).

Nocifensive movements

Group P_0_._5_, M_0_._5_ and L_0_._5_ resulted overall in less nocifensive movements during castration compared to placebo groups (S_0_._3_ and S_0_._5_) and M_0_._3_. During emasculation less nocifensive movements were seen in groups M_0_._5_ and L_0_._5_ compared to P_0_._5_ (Table 2).

Cardiorespiratory changes

Overall the fewest changes of cardiorespiratory variables were noted in groups L_0_._5_ and M_0_._5_ during castration. The difference in respiratory rate was higher in S_0_._5_ compared to L_0_._5_ (Figure 7A). Further a 10% increase in absolute values of respiratory rate was observed in M_0_._3_, S_0_._3_ and S_0_._5_ for BL-CA-CA-Max and for M_0_._3_, S_0_._3_ and S_0_._5_ for BL-CA-POST-CA. Difference in MAP was significantly lower in groups L_0_._5_ and M_0_._5_ compared to S_0_._3_, S_0_._5_ and P_0_._3_ and in group L_0_._5_ also compared to M_0_._3_ (Figure 7C). A 10% increase in absolute MAP values was observed in P_0_._3_, M_0_._3_, S_0_._3_ and S_0_._5_ from BL-CA to CA-Max and from BL-CA to POST-CA. L_0_._5_ was the only group with a decrease in heart rate during castration. There was no group with an increase in heart rate above 10% during or after castration. Difference in heart rate was lower and difference in RR-intervals were higher in groups P_0_._5_, L_0_._3_, L_0_._5_ and M_0_._5_ compared to S_0_._3_ and S_0_._5_. In groups P_0_._3_ and M_0_._3_ difference in heart rate was higher compared to group M_0_._5_ and RR-intervals shorter compared to M_0_._5_ and L_0_._5_ (Figure 8A,C). The difference in SDHR and SDRR were lowest for M_0_._5_ and L_0_._5_. Groups S_0_._3_, S_0_._5_ and M_0_._3_ had significantly higher differences in SDHR compared to M_0_._5_, L_0_._3_ and L_0_._5_ and difference in SDRR compared to M_0_._5_ and L_0_._5_ (Figure 8E,G). The difference in LF/HF ratio was lowest for L_0_._5_ and M_0_._5_ with a statistically significant difference to S_0_._3_, S_0_._5_ and M_0_._3_. In group L_0_._5_ the difference was also significantly lower compared to P_0_._3_ and L_0_._3_ (Figure 8I).

#### 3.2.4. Post-Castration Phase

Cardiorespiratory changes

Differences in respiratory rate between BL-CA and POST-CA were lower in P_0_._5_, L_0_._3_, L_0_._5_ and M_0_._5_ compared to M_0_._3_, S_0_._3_ and S_0_._5_ (Figure 7B). Groups P_0_._5_, L_0_._5_ and M_0_._5_ showed statistically significant lower differences in MAP compared to all other groups except L_0_._3_ (Figure 7D). The differences in heart rate were lower and in RR-intervals greater in groups L_0_._3_, L_0_._5_ and M_0_._5_ compared to S_0_._3_ and S_0_._5_. The difference in heart rate increased further after castration and the RR-intervals became shorter in both procaine groups.

### 3.3. Missing Data

Catheter placement for arterial blood pressure measurement was successful in Group P_0_._3_ in *n* = 14, in group P_0_._5_ in *n* = 14, in group L_0_._3_ in *n* = 14, in group L_0_._5_ in *n* = 14, in group M_0_._3_ in *n* = 16, in group M_0_._5_ in *n* = 16, in group S_0_._3_ in *n* = 16, in group S_0_._5_ in *n* = 15, and in group IM in *n* = 15 piglets. Due to technical problems and movement of piglets during injection and castration with displacement of the nasal feeding tube, measurement of respiratory rate was not determined during castration in group L_0_._5_ in *n* = 1, during exteriorization of testicles in group S_0_._3_ in *n* = 1 and in group S_0_._5_ in *n* = 1, during emasculation in group S_0_._5_ in *n* = 1, and post operatively in group P_0_._5_ in *n* = 1 as well as in one piglet in group IM during the whole procedure.

Furthermore, ECG recordings from two animals in group P_0_._3_ and data for baseline castration, castration and POST-CA^e^ in one animal in group M_0_._3_ were excluded due to technical errors and artifacts.

### 3.4. Supplementrary Material

Absolute values for respiratory rate and mean arterial blood pressure for the timepoints baseline injection (BL-INJ), injection (INJ-MAX), baseline castration (BL-CA), castration (CA) and post castration (POST-CA) as well as absolute values for heart rate, RR-Interval, SDHR, SDRR and LF/HF-ratio for the timepoints baseline injection (BL-INJ^e^), injection (INJ^e^), baseline castration (BL-CA^e^), castration (CA^e^) and post castration (POST-CA^e^) can be found in the Appendix A.

## 4. Discussion

Our main hypothesis was that local anaesthesia with mepivacaine, lidocaine or procaine will lead to significant reduction in EEG responses (SEF 95% changes) during castration. At the beginning of the examination, the NI could confirm that all piglets were in a consistent, light plane of anaesthesia across the treatment groups. However, in contrast to observations made by Waldmann et al. [21] and Haga and Ranheim [6] in pigs as well as Otto and Mally [34] and Otto [35] in sheep, we were not able to detect consistent arousal reactions in response noxious stimuli by desynchronisation or synchronisation with increase or decrease in EEG parameters. One cause may be the time-delayed display of the stages and index values of the Narcotrend^®^ monitor. Pilge et al. [36] described a time delay of 30–65 s and Klesper et al. [37] of 20–175 s. In the present study, stimuli were set in a period of 60 s to 120 s; accordingly, an almost simultaneous adjustment of the display is necessary for interpretation. In other studies in dogs [38], cattle [39] and pigs [25,40] consistent EEG changes could also not be observed after painful stimuli. Therefore, EEG indices are not further discussed. Independently of the EEG examinations, it was possible to detect the painful interventions with the help of the other variables collected. Our second and third hypothesis, that testicular injection leads to more nocifensive responses than i.m. injection and that, local anaesthesia with lidocaine and mepivacaine leads to less nocifensive and autonomic responses to castration than procaine were confirmed.

### 4.1. Sub -MAC Isoflurane Concentration for Pain Assessment

In this study, the basis for studying the pain responses of the piglets during castration was a constant, reproducible, superficial anaesthesia that prevented stress-induced responses due to handling and fixation, but still allowed nociception induced changes in vital signs to be detected, as well as nocifensive movements. It was essential to keep all piglets under the same conditions to avoid bias in the results. To quantify and standardize depth of anaesthesia, the concept of minimum alveolar concentration (MAC) is useful in animal studies because of a small inter-individual variance [41]. The MAC of inhaled anaesthetics is defined as the alveolar gas concentration at sea level required to prevent purposeful movements in 50% of patients in response to surgical incision. Age has been shown to affect MAC [42]. Therefore, MAC determined in a pre-trial in piglets of the same age group (age: 2–7 days of life) was used to define superficial anaesthesia. In this pre-trial, MAC of 1.2 ± 0.3 vol% isoflurane was determined by electric stimuli [33], a similar value (1.20 ± 0.43 vol%) was determined by the clamping technique in 2–17 day old piglets [43]. A higher MAC value of 1.41–2.00% is given by the manufacturer of the isoflurane preparation used [44]. To allow nociception assessment, end-tidal concentration was adjusted to 0.8 MAC, which corresponds to 1.0 vol% isoflurane according to our previous study [33]. The EEG recordings showed that all animals were at comparable depths of anaesthesia at baseline. Subanaesthetic isoflurane concentrations have little or no antinociceptive effect [45] and withdrawal movements triggered by noxious stimulation can be elicited at sub-MAC anaesthetic concentrations [46], while reactions due to handling are suppressed. To measure nociception of piglets during injection and castration, Saller et al. [18] chose individual MAC isoflurane anaesthesia for their minimal anaesthesia protocol determined by single reactions to a toe pinch. Although the applied mean end-tidal concentration, which was finally used, was not mentioned, it seems to be higher than the MAC we chose, as they started with 1.69 ± 0.3 vol% with a flow of 3 L/min oxygen and adapted anaesthetic depth in steps of 0.2%. Similar to our own observations they could see in blood pressure and heart rate as well as limb movements, associated with nociceptive stimuli. Haga and Ranheim [6] made corresponding observations using 1.4 x MAC of halothane anaesthesia.

### 4.2. Injection of Local Anaesthetics and Onset of Action

The local anaesthetics were injected into the testis and subscrotal tissue. The results of the preliminary study demonstrated, that the application at these sites showed a widespread distribution of a mixture of contrast medium and lidocaine beneath the scrotal skin, in the testis, and in the spermatic cord [47]. However, radiolabelled lidocaine injected into the testis did not readily diffuse through the tunica vaginalis and into the cremaster muscle. Therefore, Ranheim and Haga [48] assumed incomplete block of the sensory innervation resulting in insufficient analgesia during castration. This is confirmed by the results of our study, since regardless of the type and dosage of the local anaesthetic used, a complete absence of changes in the observed pain indicators could not be achieved, even in the M_0_._5_ group being the most effective group with the least deviations from the baseline values.

When using local anaesthetics, the maximum amount that can be applied to a piglet must be considered to avoid toxicity [9]. A wide variety of doses and concentrations of local anaesthetics used in piglet castration have been described, but toxic side effects were rarely mentioned. Nevertheless, in the present study, the recommended drug associated maximum dosages given by VETIDATA, a web-based veterinary information system (www.vetidata.de (accessed on 11 February 2020), and an easy to handle volume of commercially available products (0.5 mL) were compared. No toxic side effects were observed, despite exceeding the maximum recommended dose by a multiple when using 0.5 mL of the local anaesthetics per injection site (2 mL per piglet). Due to the close relationship between systemic toxicity and the plasma concentration of local anaesthetics [49], the maximum plasma level is decisive for the onset of side effects. Scott et al. [50] demonstrated that the maximum plasma concentration of lidocaine occurred between 10 and 20 min after injection comparing 4 injection sites (intercostal, subcutaneous vaginal, lumbar epidural, subcutaneous abdominal). During this time, however, the infiltrated testis has already been removed, so that complete absorption could not take place, which may help to protect the piglet from systemic side effects.

Comparing the median onset of action of the local anaesthetics, no significant differences except for mepivacaine in the low application volume (M_0_._3_) could be detected, M_03_ and S_03_ had the longest median with 11 min. Interestingly, quite a few animals, which got saline 0.9%, did not show any aversive reactions to pinching of the scrotal skin within the 15 control minutes. Despite the standardized procedure, the lack of reaction may have been caused by the inhalation anaesthesia or reflects the individual variation of pain perception [51].

### 4.3. Pain Assessment during Injection

We compared pain induced reactions during intratesticular/subscrotal injection with intramuscular injection, because the latter is a routine route of drug administration in pigs. Therefore, the infliction of such pain is generally accepted also by the German Animal Welfare Act. The diameter of the canula for the i.m. injections was comparable to routinely used cannulas in the field. To reduce the pain caused by needle insertion into the testis, a smaller needle was used, as recommended [52,53]. Pain associated reactions were only assessed in connection with injection of the local anaesthetic or saline solution. However, pain responses to the sole puncture of the skin and the testis might also be important in the assessment of animal welfare aspects in piglet castration.

Regardless of the localisation and the content, every injection has caused nociception, but there were marked variations in the pain response. Volunteers recognized differences after intradermal and subcutaneous infiltration of five local anaesthetics, which were not related to the acidity, ionization, protein binding, sodium chloride concentration or osmolality, but a relation to lipid solubility was assumed [54]. Accordingly, the injection of mepivacaine was perceived as more painful than lidocaine [54]. The current results indicate more influence of the injected volume than lipid solubility, since no differences were found after injection of mepivacaine or lidocaine. In fact, after smaller intratesticular and subscrotal volumes fewer pigs showed nocifensive movements compared to the control group in which the pigs received 0.5 mL saline i.m. At the same time, RR as well as MAP rose above the 10% of baseline in those groups which received a volume of 0.5 mL per injection site, regardless of the agent. An increase of 10% from baseline was defined as a positive response to the invasive intervention according to Otto et al. [55] and Roizen et al. [56]. Additionally, although not statistically significant, the pain induced reactions after the larger volume of saline led to stronger responses. Therefore, it is assumed that the pain induced reaction is caused in particular by the pressure in the tissue, as the volume of the testicles cannot expand due to the firm layers. In contrast, Coutant et al. [57] could not observe any differences in vocalisation and foreleg movements after intratesticular application of 0,3 mL or 0.5 mL procaine 2%.

According to previous studies, procaine injection causes the greatest discomfort [13,19,58]. Tissue irritation due to the significantly lower pH of procaine (pH of 3.7) compared to lidocaine (pH of 5.0) and mepivacaine (pH of 5.5) cannot be ruled out as a cause. The sensation of nociception was confirmed by the changes in the LF/HF ratio of the HRV. Increase in LF and LF/HF ratio reflects sympathetic baroreflex activity [59,60], while HF reflects parasympathetic influences and corresponds to the HR variation depending on the respiratory cycle (respiratory sinus arrhythmia) [61]. The increase in the LF/HF ratio is accompanied by an increase in MAP, one of the most sensitive nociceptive indicators in pigs [25].

The simultaneously observed increases in time domain indices (SDHR, SDRR) were unexpected as with an increased heart rate accelerated by the sympathetic nerves, time between heartbeats (RR intervals) decreased and less time for variability occur, which means, HRV should actually decrease. Additionally, the heart rate did not change as expected. We could notice a small decrease in heart rate (<3.5% from baseline) as well as an increase in RR-intervals during the minute after start of injection in the L_0_._3_, M_0_._3_ and S_0_._3_ group. Clement et al. [62] reported sudden and relatively sustained falls in both arterial pressure (up to 56% from baseline) and heart rate (up to 30% from baseline) after noxious deep somatic and noxious visceral manipulations. Haga and Ranheim [63] also observed a decrease in the pulse rate during injection of lidocaine into the funiculus spermaticus in piglets, while Saller et al. [18] reported a decrease in blood pressure and heart rate in response to the castration. This paradoxical effect is the result of a vasovagal response, whereby bradycardia is caused by a sudden increase in vagal activity and hypotension results from a sudden reduction in sympathetic activity and relaxation of arterial resistance vessels [64]. As HRV reflects the activity of the sympathetic and parasympathetic nervous systems, changes in the recorded parameters in those three groups could be expected by increasing HRV due to the prolonged RR interval. However, SDHR and SDRR increased in all groups.

Burton et al. [65] observed a temporary increase in the heart rate (7.0 ± 2.0 %) after inducing pain by intramuscular or subcutaneous injections of hypertonic saline which returned to baseline within 60 s after inducing pain reflecting only brief arousal responses, but causing an increase in the LF/HF ratio of HRV. Radeisen [31] also could relate changes in frequency domain indices to intraoperative pain associated nocifensive movements during castration of boars in the pubic region, especially during incision of the skin and the vaginal process and during the traction of the spermatic cord, whereby time domain HRV did not reflect those single pain events. Using the LF/HF ratio Raue et al. [66] could detect nociceptive stimulation in cats in only 60 s post stimulation epoch measurements, while longer time periods revealed a readjustment to the basal values. Generally, short-term measures of HRV rapidly return to baseline after transient disturbances [28], therefore the recording time has a significant influence especially on time domain values [30] and a rapid recovery of the heart rate after the injection might be the reason for the rise in SDHR and SDRR.

### 4.4. Pain Assessment during Castration

In group M_0_._5_ (*n* = 17; 85.0%) the lowest number of animals showed nocifensive movements during castration followed by group L_0_._5_ (*n* = 15; 75.0%). Even in each of the placebo groups there were two animals (10.0%) that showed no withdrawal movements during castration possibly as a result of isoflurane anaesthesia. Saller et al. [18] also noticed three out of nine piglets of a positive control group, which received sodium chloride (without pain relief) without limb movements during castration using a light isoflurane anaesthesia model.

Autonomic responses to castration were most pronounced in the placebo groups as well as in the M_0_._3_ group during castration. Due to the high dilution of mepivacaine (depending on bodyweight of the piglet 1:23–1:45), which was based on the toxicity limit for the horse, as no corresponding value was available for the pig, effective local anaesthesia could hardly be expected. An increase in RR and MAP of over 10% above the baseline value was noted, which was not seen in the other groups, so that a marked perception of pain must be assumed. The groups P_0_._5_, L_0_._5_ and M_0_._5_ differed significantly from the placebo groups in differences of MAP, HR, and RR interval recorded during castration, while only in the L_0_._5_ group the HR slightly dropped and consequently the RR intervals increase. Nevertheless, SDHR and SDRR increased according to the changes during the injection, but this could be related to the alternating influences of the parasympathetic and sympathetic nervous systems during the different steps of castration and the pain inflicted in the sequence of skin incision, exteriorization of the testicles and emasculation. Especially the influence of the parasympathetic nervous system leads to rapid changes in the range of milliseconds, while the effects of the sympathetic nervous system are rather slow (time scale of seconds) [67].

Besides the differences mentioned above, L_0_._5_ and the Mepi_0_._5_ differed significantly from the placebo groups in the differential values of the LF/HF ratio. This suggests that both substances are best at blocking sympathetic baroreflex activity, but only if it is given in the appropriate dosage.

Compared to basal levels, the M_0_._5_ group showed the least changes in HR variability parameters and therefore appears superior in efficacy to lidocaine and procaine.

### 4.5. Pain Assessment Post Castration

In the immediate postoperative phase, the values of RR and MAP in group P_0_._3_, M_0_._3_, S_0_._3_, and S_0_._5_ remained at a similar level or showed a further increase compared to the values recorded during castration, indicating ongoing nociception due to the lack of analgesia during and after castration. This observation also illustrates, that despite the timely administration of the NSAID, acute postoperative pain is not effectively eliminated. SDHR, SDRR and LF/RF ratio returned to baseline levels from before castration in all groups, according to the statement of von Borell et al. [28], that short-term measures of HRV rapidly return to baseline after transient distress. Overall, during castration, the lowest deviations in autonomous responses from the baseline values were observed in the L_0_._5_ and M_0_._5_ groups, so that the most effective pain reduction is also assumed for the postoperative phase with these two agents.

### 4.6. Limitations

Limitations of this study include the change in administered volume and dose between the 0.3 and 0.5 groups so that the individual effects of either dose or volume cannot be differentiated between treatments. Isoflurane was used to facilitate immobility of the piglets. As it influenced electrical brain activity it may have blunted the EEG parameters despite the very low concentration of isoflurane used. Moreover, arterial cannulisation was not possible in all piglets so that data for MAP were not available for all piglets.

## 5. Conclusions

Contrary to our hypotheses, the EEG-Parameters NI, P_TOT_, MF, and SEF_95_ did not show any changes in connection with the painful procedures and are therefore not assessed as suitable for pain detection during injection or castration of piglets. In contrast, the concordance of changes in the LF/HF ratio with the further pain indicators collected demonstrates the applicability and utility of frequency-based HRV for the detection of pain responses during suckling piglet castration.

In comparison to the intramuscular injection, more pronounced pain responses in piglets were observed with intratesticular application of the larger volume of 0.5 mL of local anaesthetics as well as saline than with the use of 0.3 mL, although regular statistical evidence is lacking.

Pain-related responses during castration were reduced using the higher volume and thus the higher content of mepivacaine and lidocaine, while procaine appears unsuitable for local anaesthesia in piglet castration

The agreement regarding pain assessment with a combination of several physiologic variables and behavioural observations leads to the conclusion that a completely pain-free castration is not feasible by means of intratesticular and subscrotal injection of procaine, lidocaine or mepivacaine in the applied concentration of 2%. Whether a higher concentration of mepivacaine and lidocaine, with a reduced volume, would lead to sufficient distribution in the tissues and complete analgesia during castration needs to be investigated in further studies.

## Figures and Tables

**Figure 1 animals-12-02833-f001:**
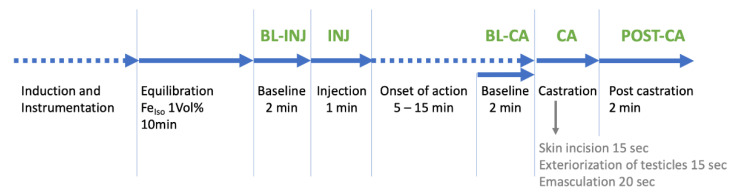
Timeline of the experimental procedure. Dotted arrows indicate time segments with variable length. Abbreviations of analyzed timepoints are green.

**Figure 2 animals-12-02833-f002:**
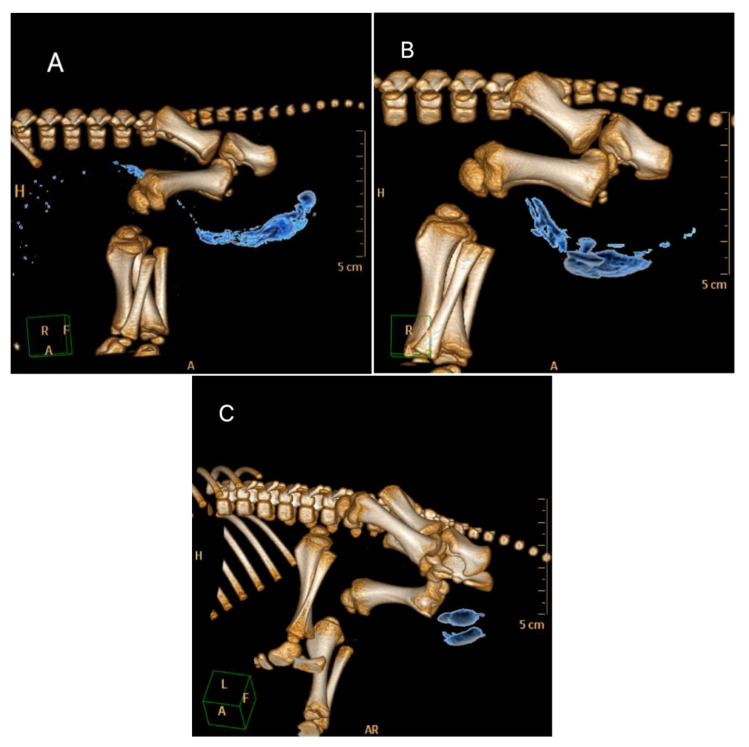
3D-reconstruction of the CT-images showing the distribution of the surrogate after intratesticular (**A**), intrafunicular (**B**) and subscrotal (**C**) injection. Orange capital letters: A-anterior, L-left, R-right, H-head, F-feet.

**Figure 3 animals-12-02833-f003:**
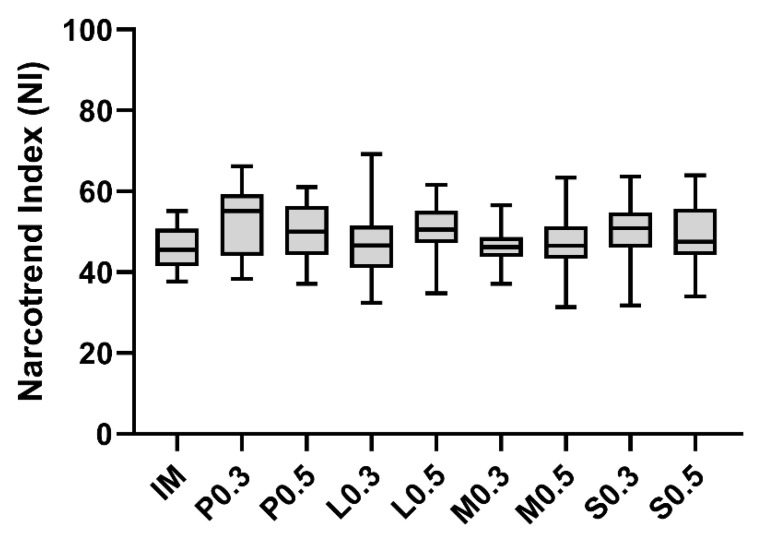
Baseline Narcotrend Index. Baseline before injection of Narcotrend Index (NI) for groups P_0_._3_, P_0_._5_, L_0_._3_, L_0_._5_, M_0_._3_, M_0_._5_, S_0_._3_, S_0_._5_ and IM. The box covers the interquartile interval, where 50% of the data are found. The whiskers indicate minimum and maximum and the band inside the box represents the median.

**Figure 4 animals-12-02833-f004:**
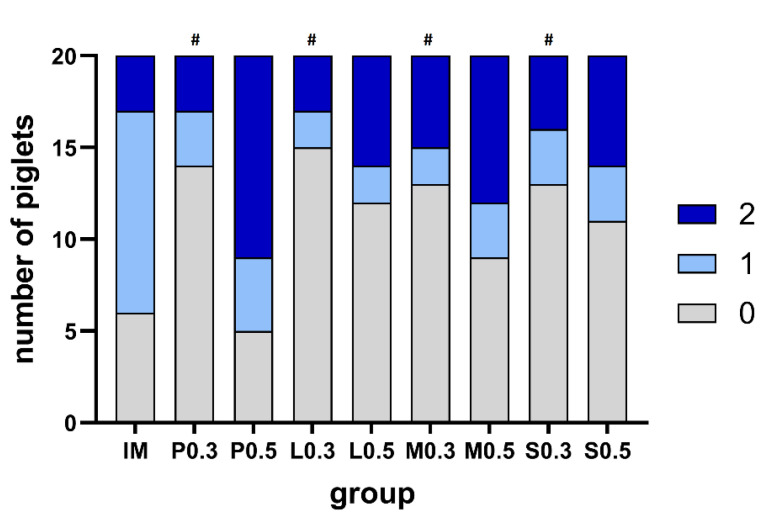
Nocifensive movements to intratesticular injection for the groups IM, P_0_._3_, P_0_._5_, L_0_._3_, L_0_._5_, M_0_._3_, M_0_._5_, S_0_._3_ and S_0_._5_. Number of piglets with a score of 0 are displayed in grey, a score of 1 in light blue and a score of 2 in dark blue. The # represents a significant difference compared to group IM.

**Figure 5 animals-12-02833-f005:**
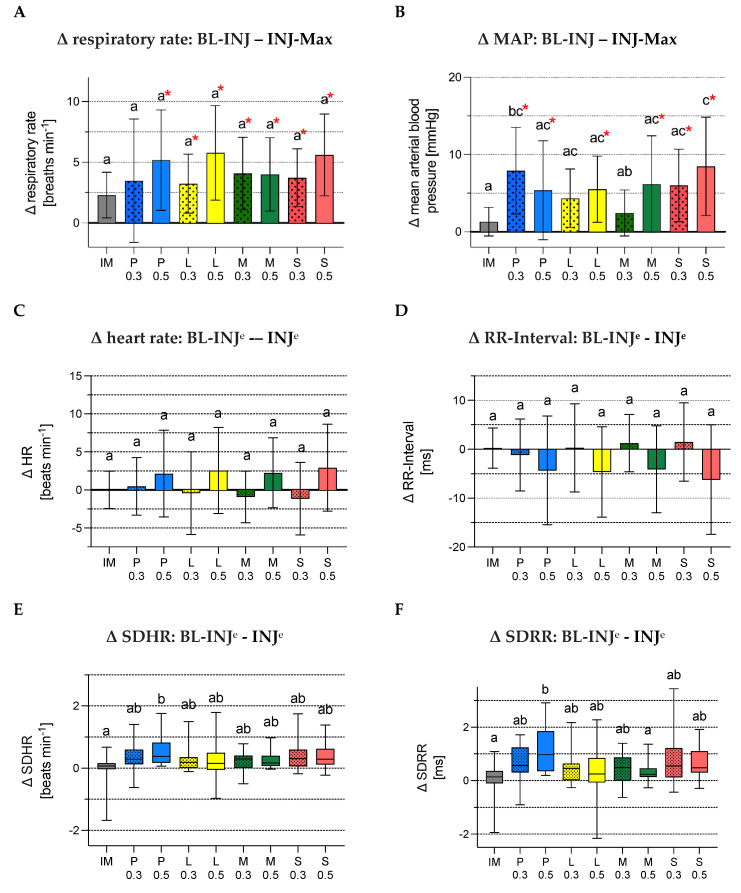
(**A**–**G**) Difference (Δ) in respiratory rate, mean arterial blood pressure, heart rate, RR-Interval, SDHR, SDRR and LF/HF-ratio between baseline and injection in the experimental groups. Bar diagrams of (**A**) mean difference in respiratory rate [breaths minute^−1^] between baseline and maximum respiratory rate during injection, (**B**) mean difference in mean arterial blood pressure (MAP) [mmHg] between baseline and maximum MAP during injection, (**C**) mean difference in heart rate [beats minute^−1^] between baseline and during injection, (**D**) mean difference in RR-Interval [ms] between baseline and during injection with the mean at the top of the bar and the whiskers as standard deviation. Boxplots of (**E**) difference in standard deviation of heart rate (SDHR) [beats minute^−1^] between baseline and during injection, (**F**) difference in standard deviation of RR-Interval (SDRR) [ms] between baseline and during injection and (**G**) difference in low frequency/high frequency (LF/HF) ratio between baseline and during injection. The boxes representing the first and third quartile and the whiskers ranging from minimum to maximum. The median is indicated by the band inside the box. Different letters (a, b, c) show significant differences between the experimental groups (*p* < 0.05). The superscript “^e”^ differentiates time periods/episodes(“^e^”) from time points (MAP and respiratory rate). The red asterisk (*) marks an increase above 10% from baseline.

**Figure 6 animals-12-02833-f006:**
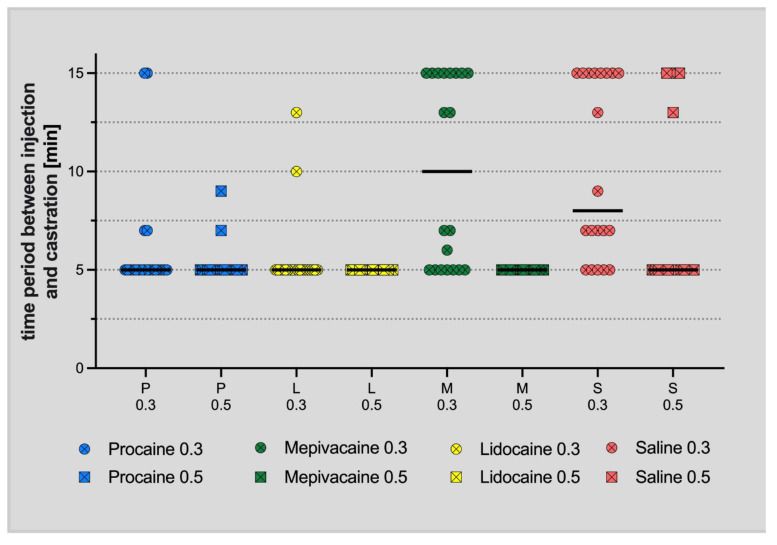
Scatter dot plot of the time period in min between the intratesticular/subscrotal application and castration for groups P_0_._3_, P_0_._5_, L_0_._3_, L_0_._5_, M_0_._3_, M_0_._5_, S_0_._3_ and S_0_._5_. The black bar represents the median.

**Figure 7 animals-12-02833-f007:**
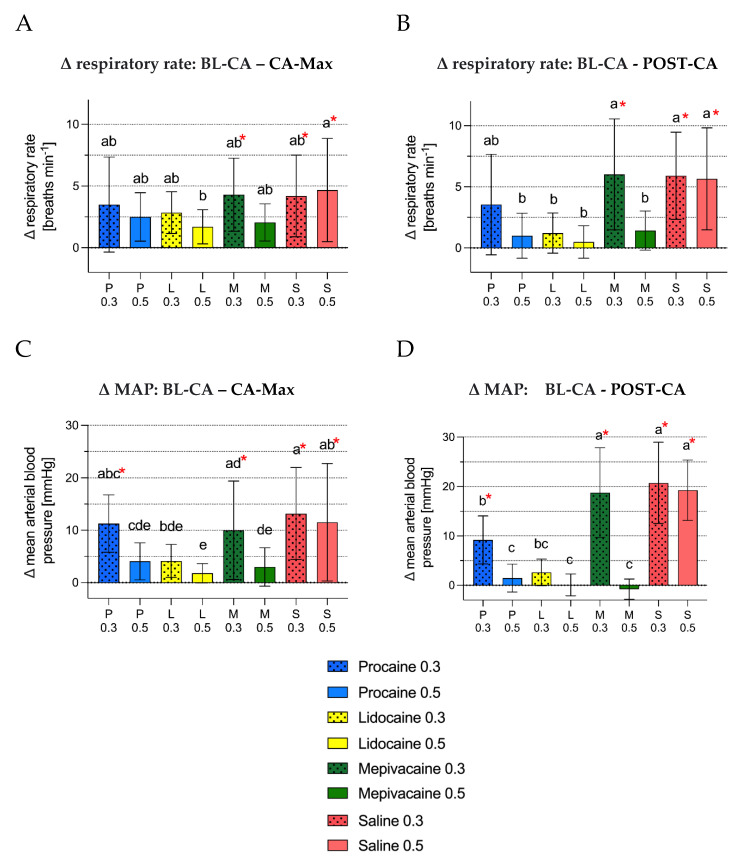
(**A**–**D**) Mean (SD) difference (Δ) in respiratory rate and mean arterial blood pressure between baseline and during castration and between baseline and post castration in the experimental groups. Bar diagrams of (**A**) mean difference (Δ) in respiratory rate [breaths minute^−1^] between baseline and maximum respiratory rate during castration and (**B**) between baseline and post castration as well as bar diagrams of (**C**) mean difference in mean arterial blood pressure (MAP) [mmHg] between baseline and maximum MAP during castration and (**D**) between baseline and post castration. The bar indicates the mean and standard deviation is represented by the whiskers. Different letters (a, b, c, d, e) show significant differences between the experimental groups (*p* < 0.05). The red asterisk (*) marks an increase above 10% from baseline.

**Figure 8 animals-12-02833-f008:**
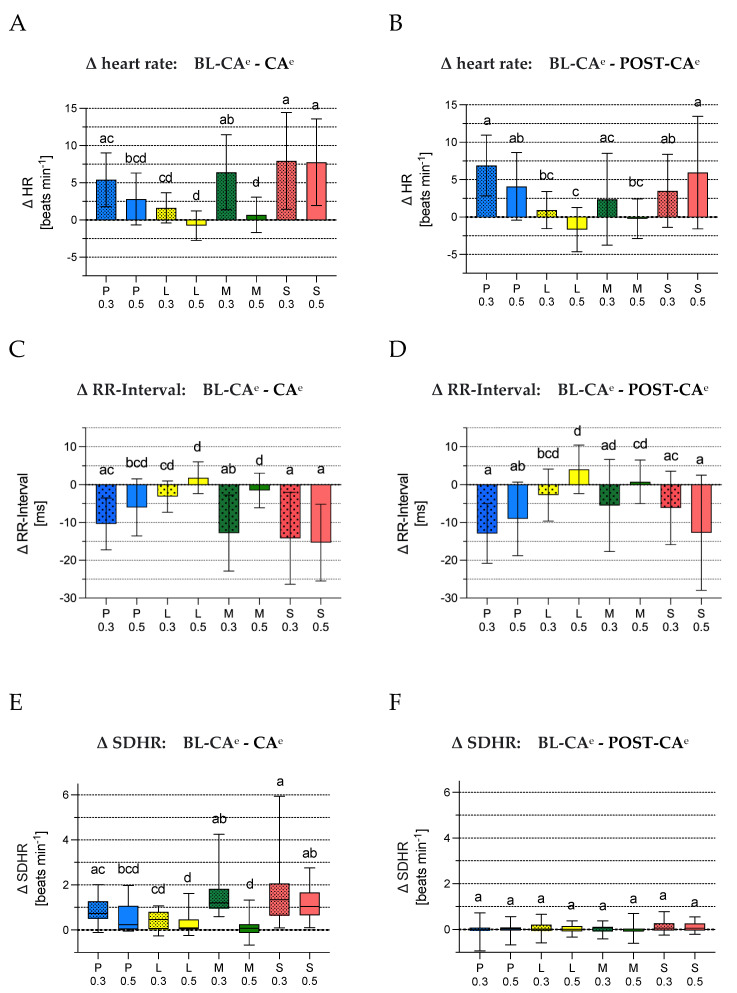
(**A**-**J**) Mean difference in heart rate, RR-Interval, SDHR, SDRR and LF/HF-ratio between baseline and during castration and between baseline and post castration in the experimental groups. Bar diagrams of (**A**) mean difference in heart rate [beats minute^−1^] between baseline and during castration and (**B**) between baseline and post castration as well as bar diagrams of (**C**) mean difference in RR-Interval [ms] between baseline and during castration and (**D**) between baseline and post castration. The bar indicates the mean is displayed as the top of the bar and standard deviation is represented by the whiskers. Boxplots of (**E**) difference in standard deviation of heart rate (SDHR) [beats minute^−1^] between baseline and during castration and (**F**) between baseline and post castration, (**G**) difference in standard deviation of RR-Interval (SDRR) [ms] between baseline and during castration and (**H**) between baseline and post castration and (**I**) difference in low frequency/high frequency (LF/HF) ratio between baseline and during castration and (**J**) between baseline and post castration. The boxes represent the first and third quartile and the whiskers range from minimum to maximum. The median is indicated by the band inside the box. Different letters (a, b, c, d) show significant differences between the experimental groups (*p* < 0.05). The superscript “^e”^ differentiates time periods/episodes(“^e^”) from time points (MAP and respiratory rate).

**Table 1 animals-12-02833-t001:** Treatment groups, injected volume, total dose and concentration of the local anaesthetic.

Group	P_0_._3_	P_0_._5_	L_0_._3_	L_0_._5_	M_0_._3_	M_0_._5_	S_0_._3_	S_0_._5_	IM
drug	procaine 2%	procaine 2%	lidocaine 2%	lidocaine 2%	mepivacaine 2%	mepivacaine 2%	saline 0.9%	saline 0.9%	saline 0.9%
volume per site (mL)	0.3	0.5	0.3	0.5	0.3	0.5	0.3	0.5	0.5
total dose	5 mg kg^−1^ diluted in saline	40 mg undiluted	4 mg kg^−1^ diluted in saline	40 mg undiluted	0.4 mg kg^−1^ diluted in saline	40 mg undiluted			
total volume (mL)	1.2	2	1.2	2	1.2	2	1.2	2	0.5

**Table 2 animals-12-02833-t002:** Number of piglets showing nocifensive movements during skin incision, exteriorization of testicles and emasculation (*n* = 20/group).

	P_0_._3_	P_0_._5_	L_0_._3_	L_0_._5_	M_0_._3_	M_0_._5_	S_0_._3_	S_0_._5_
skin incision	6 ^a,c^	1 ^a^	9 ^b,c^	4 ^a^	12 ^c,d^	1 ^a^	10 ^c,d^	14 ^d^
exteriorization	10 ^b^	1 ^a^	4 ^a^	2 ^a^	16 ^c^	1 ^a^	18 ^c^	15 ^b,c^
emasculation	16 ^c^	8 ^b^	9 ^b^	1 ^a^	19 ^c^	3 ^a^	18 ^c^	17 ^c^
no reaction	2	10	6	15	0	17	2	2

^a, b, c, d^: Different letters (a, b, c, d) show significant differences between the experimental groups (*p* < 0.05).

## Data Availability

The data presented in this study are available in Appendix A. Additional data are available on request from the corresponding authors.

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
