# Peer review of "Comparative Study of Pain-Related Responses of Male Piglets up to Seven Days of Age to the Application of Different Local Anaesthetics and Subsequent Castration"

_animals, 2022, doi:10.3390/ani12202833_

Round 1

Reviewer 1 Report

This manuscript provides information on the efficacy of three different local anaesthetics (Procaine, Lidocaine, Mepivacaine) for pain elimination during the surgical castration of male piglets and the suitability of different physiological variables for pain assessment. The results of the study underline the challenging interpretation of EEG results and outlines its limits. Furthermore, comprehensive data of heart rate variability parameters are provided as well as data on the analgesia of the skin, injection, and postoperative pain evaluation.

This is an interesting paper addressing some parameters for pain assessment during the surgical castration of male piglets that have not been assessed a lot before, like EEG or heart rate variability. Also, the evaluation of the analgesia of the skin and the postoperative pain contributes valuable information to the complex discussion on the feasibility of local anaesthesia for suckling piglet castration. However, there are some weaknesses especially in the structure and presentation of the methods and results part. In general, the spelling and sometimes the language style should be revised, there are a few commas missing. Also, a list of abbreviations would be helpful. The literature should be reviewed for current papers that should be included in the discussion.

L 91: the standard abbreviation is EMA

Material and methods

In general, the description of the materials and methods misses some details, please include all information so that the results are reproducible. The superscriptions should be revised (for example statistical analysis should be 2.3) and the structure adapted (Injection technique, nocifensive movements, castration, data recording).

L 145 ff: more information on the animals should be provided as they were acquired from four different farms: how many animals from which farm, were they equally subdivided in the groups? Did you analyze if there was a farm effect on the results?

L 158: Pre-trial: please describe the injection techniques more detailed: what was the length of the cannulas? What was the exact procedure of the five injection techniques? How did you make sure it was an intrafunicular injection?

L 191: It should be clarified that the dose differed, too. The groups with 0.5 ml cannot be compared with the 0.3 ml groups as the volume was reduced at the same time with the dose of the local anaesthetic. Also, it is not clear if the IM animals were castrated (as a negative control group?).

L 209: Is that right that the piglets for the castration study were not intubated and if yes, why?

L 231 ff: Again, please describe the injection techniques more detailed: was aspirated before administering the local anaesthetic?

L 243 ff: What is a purposeful movement exactly, how was it differentiated from a not purposeful movement? When an animal showed a non-purposeful movement, was it counted to score 0 = no movement? Was it the same person scoring all the animals and was it scored in the moment of the procedure or was it captured on video?

L 247 ff: Please describe the clamping more detailed: what was the location, how long and with which pressure and which clamp was it performed with? Why were the animals clamped one minute after injection when the castration started at the earliest 5 minutes after injection? Is it possible that some animals were kept awake by the repeated stimulus?

L 253 ff: please describe the castration more detailed: were both testicles removed together? Was the Meloxicam administered 20 minutes before the castration or before injection?

L 258: “…during emasculation the emasculator was …”

L 267, 278: An abbreviation should only be used for one specific expression. For example, „BL-CA“ describes the first time when it appears a time point of two parameters (respiratory rate and blood pressure), later the same abbreviation “BL-CA” stands for a different parameter describing a period of time (heart variability parameters, ECG).

L 302: Figure 1 is not clear, for a better understanding it would be helpful to include the abbreviations that are explained in the text describing the time points / periods of time as well as the parameters. In the text describing the heart rate variability parameters, “CA” includes start of skin incision until 20 seconds after start of emasculation, in figure 1 it says skin incision 15 seconds.

Results:

L 336: To structure the results part, it would make sense to name 3.3. “main study” and order the EEG results under Response to Injection and Response to Castration, respectively.

L 341: I don’t see the additional information given in Figure 3, it is described in the text (l 337 ff).

L 354: Figure 4: was the difference between the four 0.5 groups and the 0.3 groups calculated?

L 360: More information on these data would be interesting, for example a scatter plot or a boxplot to visualize the variance of the data.

L 367: This sentence is a unclear, it must be clarified that there was no significant difference between the groups in the respiratory rate. In the suppl. material it is shown that only between BL-INJ and INJ-Max there was a significant difference in the respiratory rate.

Discussion

Again, the discussion is not easy to understand due to the structure of the superscriptions. In lines 533 ff the pain after injection is discussed and later the passage in lines 568 ff has the superscription “pain assessment during injection”.

It would be interesting to discuss the results of the “onset of action”. I think the analgesia of the skin is an important and interesting point, but I could not find anything about these results in the discussion. How do you interpret the median of 5 for the S0.5 group?

L 579: please describe the 10% above basal level in the methods and results. How do you interpret that in the nocifensive movements to injection the P0.5 group showed significantly more reactions when in the MAP BL-INJ – INJ-Max there was a significant difference between P0.3 group and IM but not P0.5 and IM (and the other 0.5-groups).

Conclusion

The acceptance or rejection of the main hypotheses should be included in the conclusion.

L 681 ff: The conclusion that the injection of a lower concentration is less painful can’t be drawn because it is not clear if the results can be explained by only the differing volume or additionally by the lower concentration of the local anaesthetic.

Supplementary Data

L 723: it is interesting that there is a statistically significant difference in the values of the respiratory rate in L0.5 during castration compared to the baseline.

Author Response

Dear reviewer,

thank you for your time and expertise you put forward to improve this manuscript. We really appreciate your effort and we have tried to address all of your concerns and comments. Our comments on your questions are written in blue.

Thank you very much

Reviewer 2 Report

The study turns out to be interesting and is very actually topic in veterinary welfare.

Unfortunately, there are several spelling errors that give the idea of not having cured the shape much. Errors are also present in acronyms, tools and drugs style. Style should be standardized.

Some sentences are difficult to understand, this may affect the quality of the manuscript.

Most tables are clear and understandable.

Given the large amount of data processed, the discussions could be expanded.

References must be rechecked and standardized in the stylle.

Author Response

Thank you very much for your thorough review and pointing out incorrect spelling and language errors.

We have tried to make all necessary corrections, introduce acronyms correctly and use them in a consistent way. An intensive revision of the English style and expression has also been done throughout the manuscript using track changes. We have also tried to answer the majority of your comments in the file with your original comments.

Thanks again for helping to improve our manuscript.

Round 2

Reviewer 2 Report

As mentioned in the first round, the manuscript is original and veri interesting. The manuscript has been substantially revised without affecting the design of the study. The attention given to revision makes the manuscript much smoother and more enjoyable to read. The comments are exhaustive.

Thank you for your comments

Best regards